# Cross Talk of Macrophages with Tumor Microenvironment Cells and Modulation of Macrophages in Cancer by Virotherapy

**DOI:** 10.3390/biomedicines9101309

**Published:** 2021-09-24

**Authors:** Sarah Di Somma, Fabiana Napolitano, Giuseppe Portella, Anna Maria Malfitano

**Affiliations:** Dipartimento di Scienze Mediche Traslazionali, Università di Napoli Federico II, 80131 Napoli, Italy; sarah.ds@libero.it (S.D.S.); fabiananapolitano94@gmail.com (F.N.)

**Keywords:** macrophages, tumor microenvironment, cancer, mesenchymal stem cells, fibroblasts, virotherapy

## Abstract

Cellular compartments constituting the tumor microenvironment including immune cells, fibroblasts, endothelial cells, and mesenchymal stromal/stem cells communicate with malignant cells to orchestrate a series of signals that contribute to the evolution of the tumor microenvironment. In this study, we will focus on the interplay in tumor microenvironment between macrophages and mesenchymal stem cells and macrophages and fibroblasts. In particular, cell–cell interaction and mediators secreted by these cells will be examined to explain pro/anti-tumor phenotypes induced in macrophages. Nonetheless, in the context of virotherapy, the response of macrophages as a consequence of treatment with oncolytic viruses will be analyzed regarding their polarization status and their pro/anti-tumor response.

## 1. Introduction

The tumor microenvironment (TME) is constituted by several components: neoplastic cells, fibroblasts, endothelial cells, macrophages, lymphocytes, and mesenchymal stem cells. The release of cytokines, chemokines and other proteins and factors by these cells contributes to the creation of a pro-tumorigenic TME. The interplays among these cells via cell–cell contact or secreted factors, represent key processes orchestrated by signals of neoplastic cells, that induce cell proliferation, migration, differentiation, angiogenesis, immune cell recruitment. The microenvironment of growing tumors is characterized by the co-existence of proliferating cells and dying cells that co-operate to define aberrant changes of normal tissues. Dying cells release danger signals re-calling macrophages that in the attempt to establish tissue-repairing mechanisms, promote vascularization, removal of cell debris, and immunosuppression, all conditions leading to the acquisition of pro-tumorigenic features of macrophages [1]. In this context, the cross talk with cancer and other cells of the TME is relevant to orchestrate the macrophage polarization status. In this review, we will focus on the cross talk between macrophages and the main cells of TME contributing to define their polarization profile, like cancer cells, mesenchymal stem cells (MSCs), and fibroblasts that can acquire a cancer-associated fibroblast (CAF) phenotype.

Among the experimental but also clinically approved therapeutic approaches adopted against cancer, virotherapy represents a cancer treatment based on the use of native or reprogrammed viruses, the oncolytic viruses (OVs) endowed with the potential to target cancer cells. In particular, OVs infect and replicate specifically in cancer cells determining their lyses [2]. Additionally, OVs act via a multiple of mechanisms that modulate TME, including the induction of immunogenic cell death (ICD) with the activation of danger signals that stimulate the immune response against tumors [3]. The successful outcome of virotherapy is supported by the increasing number of clinical trials and by the recent approval of OVs for the treatment of several types of cancers. Among these OVs, Rigvir is a wild type non-pathogenic ECHO virus type 7 (ECHO-7) registered for the treatment of melanoma in Latvia, Armenia, Georgia, and Uzbekistan; H101 is an attenuated adenovirus bearing the same deletion of dl1520 approved in 2005 by China’s State Food and Drug administration (FDA) for the treatment of head and neck carcinoma; T-VEC (talimogene laherparepvec) is a modified herpes simplex virus type-1 (HSV)-1 expressing GM-CSF approved in 2015, by both the European Medical Agency (EMA) and FDA [4]. Nonetheless, combinatory therapeutic approaches using OVs with other anticancer agents [4,5,6,7] were suggested to enhance oncolytic activity and stimulate a robust antitumor immune response. We will here evaluate the effect of OVs on TME modification, focusing on macrophage phenotype changes.

## 2. Macrophage–Cancer Cell Cross Talk and TME Remodeling

In solid tumors, macrophages represent more than 50% of the tumor mass and tumor-associated macrophages (TAMs) can develop from circulating precursor monocytes and resident macrophages. Environmental perturbations make TAMs susceptible to the acquisition of a wide spectrum of metabolic, functional, and phenotypic profiles ranging from the M1 to the M2 polarization. Upon activation, M1 macrophages secrete pro-inflammatory cytokines and express specific markers (MHC-II, CD68, CD80, CD86, pSTAT, and iNOS) similar to macrophages stimulated by microbial substrates, i.e., lipopolysaccharide (LPS), toll-like receptor (TLR) ligands, and Th1 cytokines. In particular, the secretion of pro-inflammatory cytokines is related to several transcription factors, i.e., STAT1 that induces numerous inflammatory molecules. STAT1 represents one of the most important transcription factors involved in the M1 polarization of macrophages. Upon phosphorylation, STAT1 nuclear translocation induces the expression of genes like CXCL9 related to M1 polarization [8]. M1 TAMs have a key role in host defense and cancer cell killing. M2 TAMs are stimulated by transforming growth factor β (TGFβ), chemokines, Th2 cytokines, and prostaglandin E2 (PGE2). M2 TAMs produce TGFβ, anti-inflammatory cytokines (IL-4, IL-13, IL-10) and PGE2 and express CD204, CD206, CD163, VEGF, cMAF, CD200R, macrophage galactose-type lectin (MGL) 1, and MGL2.

M2 phenotype is involved in the resolution of the inflammatory response, suppression of the immunity against parasites and tumor cells, enabling the TME to promote cancer progression and metastasis. The M2 phenotype is also characterized by upregulated genes, i.e., arginase 1 (ARG1), resistin-like molecule α (FIZZ1), chitinase-like protein Ym1, and macrophage mannose receptor (MMR) 1, MRC1 [9].

### 2.1. Cancer Cell Signals Reprogram Macrophage Function and Polarization

Tumor-derived signals induce alterations in monocytes reaching cancer sites able to drive their phenotype. In particular, cancer-immune signals, cancer metabolism, and cancer cell death can orchestrate a reprogramming process driving to the acquisition of specific TAM features [1,10].

Cancer cells are a main source of immune signals that control TAM function. Human cancer cell lines secrete cytokines able to impair macrophage anti-tumor properties, i.e., IL-10, TGF-β, and PGE2 stimulate myelopoiesis, producing granulocyte and macrophage growth factors, i.e., M-CSF (CSF-1), GM-CSF (CSF-2), and G-CSF (CSF-3). Additionally, cancer cells promote monocyte recruitment through the release of chemotactic signals such as VEGFA, CCL2, CXCL12, semaphorin 3A (SEMA3A) [9,11].

Cancer metabolism affects macrophage function, the presence of abnormal proliferating cells promotes a hypoxic TME with high concentration of lactate that up-regulates in macrophages pro-tumor genes VEGF and ARG1, representative of a M2 TAM profile.

Spontaneous or therapy-induced cancer cell death promotes the release in the TME of danger signals favoring ICD, a kind of cell death represented by ATP release, exposure of calreticulin on cancer cell surface and extracellular release of high mobility group box 1 (HMGB1) [3,12]. TAMs recognize these signals and stimulate in turn pro-inflammatory anti-tumor responses with the upregulation of an M1 profile.

### 2.2. Macrophage Signals to Cancer Cells

A bidirectional relationship governs macrophage–cancer cell interplay; macrophages are plastic cells that according to the environmental signals engage different responses, and in some contexts, elicit anti-tumor response; however, we here revise their contribution to tumor growth based on immune and non-immune mechanisms.

TAM pathogenic activity is mediated by the suppression of anticancer immune response, TAMs secrete immunosuppressive factors, i.e., IL-10, TGF-β, PGE2 that via CCL22 recruit T regulatory cells (Treg) [13]. IL-10 through a negative regulation of IL-12 secreted by myeloid cells indirectly induces Th2 cell differentiation and the release of IL-4 and IL-13, further promoting M2 TAM phenotype [14]. Additionally, IL-4, IL-10, and hypoxia-induced ARG1 control arginine catabolism and impair T cell function [15]. Arginine as the inducible nitric oxide synthase (iNOS2) substrate is required to induce nitric oxide (NO), a critical mediator of macrophage cytotoxicity, which in tumors suppresses T cell function. Indeed, the secretion of pro-inflammatory cytokines IL-6, IL-11, TNFα, and IL-1β enhances tumor cell proliferation through the signal transducer and activator of transcription 3 (STAT3) and the anti-apoptotic nuclear factor κB (NF-κB) [16].

Beyond these effects, TAMs exert a metabolic influence on cancer cells. Aberrant metabolism is a hallmark of cancer, in particular, aerobic glycolysis favors a low pH microenvironment that supports cancer cell proliferation, additionally, the end-product of this process, lactic acid, fuels cancer cells lacking energy supplies. In this context, hypoxia and lactate drive TAMs to secrete cytokines like IL-6, TNF, CCL5, and CCL18, which favor glycolysis and glycolysis factors including phosphoglycerate kinase 1 (PGK1), HXK2, lactate dehydrogenase (LDH), glucose-6-phosphate dehydrogenase (G6PD), GLUT1, vascular cell adhesion molecule 1 (VCAM1). In particular, VCAM1 further supports glycolysis, glucose uptake, and lactate secretion promoting M2 polarization as described in pancreatic cancer cells [17,18,19]. Additionally, hypoxia inducible factor 1 subunit α (HIF1A) is involved in mechanisms that via TAM-derived factors exacerbate malignant cell glycolysis and the capacity of lactate derived from cancer cells to promote M2 polarization. On the other hand, M2 TAMs actively contribute to hypoxia partly via AMP-activated protein kinase (AMPK) and PPARGC1A favoring highly oxidative environment with enhanced mitochondrial mass and increased oxygen consumption [1].

## 3. Macrophage–MSC Cross Talk in Cancer and TME Remodeling

MSCs are pluripotent stem cells isolated initially form bone marrow, but also from other tissues defined by specific characteristics: positivity of CD105, CD73, CD90 markers; low expression of HLA-I; negative expression of HLA-II, CD14, CD11b, CD45, CD34, and CD31. These cells are able to elicit osteogenic, adipogenic, and chondrogenic differentiation capacity [20]. MSC via mechanisms involving cell–cell contact and secreted molecules are able to regulate innate and adaptive immune cell functions. This regulation happens in healthy tissues and in cancer tissues; we will here focus on mechanisms observed in tumor-derived MSC. It was reported that cancer cell–MSC contact and factors secreted by transformed cells affect MSC functions. For example, breast cancer cells enhance the expression of RANTES/CCL5 in MSCs, an event that rapidly occurs via cell–cell contact [21]. Consequently, the monocyte attracting chemokine CCL5, in breast cancer but also in other tumors, enhances the presence of deleterious TAMs. Additionally, the contact bone marrow MSC-leukemic cell-VLA-4 and VCAM-I mediated increases factors contributing to cancer cell survival like IL-8, IL-6, VCAM-1, and CCL2 [22], known to correlate with high infiltration of macrophages promoting cancer cell growth.

Secreted factors in conditioned medium from head and neck squamous cell lines promote the expression of CD54/ICAM-1 in MSCs, an event that is relevant in the interplay among MSCs, immune cells, and cancer cells [23]. We here examine the interplay between MSCs and macrophages upon MSC recalling to tumor sites.

### Macrophage–MSC Interaction

MSCs migrate to the site of cancer development characterized by a pro-inflammatory microenvironment, the secretion of pro-inflammatory cytokines like IFN-γ and TNF-α enhances in MSCs the expression of cytokines, enzymes, and growth factors that through immunosuppression promote cancer progression. These cytokines are particularly relevant in the interaction between macrophages and MSCs in cancer development. TNF-α-activated MSCs recruit monocytes and macrophages via CCR2 secretion [24]. Indeed, M1 macrophages favor MSC immunosuppressive phenotypes, enhancing the expression of mRNA iNOS, COX2, IL-6, and MCP1/CCL2. In contrast, IL-6 secretion by MSCs splits M1 into M2 phenotype with increased expression of CD206 and ARG1 [25] (Figure 1). In a model of breast cancer, the secretion of IL-1β and TNF-α by transformed cells and immune cells, activates in bone marrow MSCs NF-κB pathway and chemokine secretion like CCL2 and CXCL8/IL-8 that favor an inflammatory TME promoting monocyte migration [26].

Immunosuppressive effect of MSCs on other immune cells are also reported such as on T lymphocytes, NK cells [20], but will not be examined in this review.

## 4. Macrophage–Cancer-Associated Fibroblast (CAF) Cross Talk and TME Remodeling

The origin of CAFs is still controversial; however, several cell types and precursors are believed to drive an activation state leading to CAF development. Among these cells, resident fibroblasts, hematopoietic stem cells, bone marrow-derived MSCs, and endothelial and epithelial cells are considered CAF precursors. Resident fibroblast activation is responsible for a large proportion of CAFs [27] and resident and recruited MSCs are known to differentiate and acquire a CAF phenotype that affects cancer cell proliferation and invasion capacity by secretion of cytokines, growth factors, and metabolites. CAF–cancer cell cross talk has been considered the most important mutual interaction in TME [28]. Cancer cells secrete factors, i.e., chemokines, interleukins, growth factors, and proteinases that enhance CAF activation and reactivity. Additionally, CAF presence in peripheral blood of cancer patients correlates with tumor progression and poor prognosis, indeed, the aggregates CAFs-circulating tumor cells in blood sample from patients indicate worse clinical outcomes [29]. CAFs also increase circulating tumor cells when travelling in clusters with immune cells, macrophages, and platelets. Among the interaction of CAFs with immune cells, we will focus on CAF–macrophage interplay.

### Macrophage–CAF Interaction

Several studies demonstrated that the interplay of CAFs–macrophages promotes cancer progression (Table 1) and clinical-pathological reports suggest a synergic association of CAFs and TAMs with poor prognosis. On the other hand, aggressive phenotypes including clinical stage correlate with high number of macrophages and high areas of CAFs [8]. CAFs interact with M2 macrophages to drive malignant tumor progression and their presence is often detected in the same area of tumor tissues in a proportion dependent on the type of tumor. Lymphoma, brain, kidney, and hepatocellular carcinoma tissues showed higher density of TAMs, whereas in lung pancreatic, gastrointestinal, and prostatic tissues, CAFs were detected at higher density than TAMs [30]. Studies on the influence of macrophages on CAFs are limited; however, it is known that M2 macrophages affect the epithelial-mesenchymal transition (EMT) of fibroblasts converting healthy fibroblasts into CAFs. Upon macrophage activation, fibroblasts stimulate prostatic cancer with the involvement of IL-6 and stromal cell-derived factor-1 (SDF-1) [31]. MSCs derived from human umbilical cord in co-culture with macrophages differentiate into CAF promoting gastric epithelium cell malignancy via EMT [32].

On the other hand, CAFs–cancer cells reshape the microenvironment to which monocytes are recruited to favor tumor progression. Several factors released by CAFs and tumor cells might be the main determinants of macrophage polarization and function during the differentiation of circulating monocytes. CAFs showed increased levels of IL-6, CXCL8, and TGF1 [33] and the inflammatory cytokines, IL-6 and CXCL8, are known to promote M2 macrophage polarization. CAF-recruiting monocytes via SDF-1 and monocyte chemotactic protein-1 (MCP-1) enable monocyte differentiation into M2 macrophages with increased levels of IL-10, driving their immunosuppressing role in breast cancer [34]. In high-risk neuroblastoma, CAFs produce pro-inflammatory lipid mediators that favor tumor growth and a high infiltration of CD163+ macrophages [35]. In prostate cancer, the interplay cancer cells, M2 macrophages, and CAFs enhance tumor cell motility promoting metastatic diffusion and these effects can be regulated by estrogen receptor-α (ER-α), which if highly expressed in CAFs, decreased the levels of IL-6 and CCL5 in both CAFs and macrophages co-cultured in conditioned medium [36]. Human colorectal cancer-derived CAFs can drive the adhesion of monocytes by up-regulating VCAM-1 levels in colorectal cancer cells. Additionally, CAFs can attract monocytes by secreting IL-8 rather than SDF-1 promoting M2 polarization that synergize with CAFs in suppressing natural killer (NK) cell function [37]. In hepatocellular carcinoma, CAFs and cancer cells induced the M2 polarization by upregulating the mRNA levels of CD206 and CD163, and downregulating in the macrophages IL-6 mRNA expression and secretion [38].

**Table 1 biomedicines-09-01309-t001:** Macrophage and CAF interaction in different tumors.

Type of Tumor	Macrophage/CAF Interaction	Ref.
Prostate cancer	➢ Density of CAFs > TAMs ➢ Fibroblasts induce the increase of IL-6 and SDF-1 ➢ M2 macrophages and CAFs induce tumor cell motility by ER-α with a decrease of IL-6 and CCL5	[30,36]
Breast cancer	➢ CAFs recruit monocytes via SDF-1 and MCP-1 and induce M2 macrophage polarization with increase of IL-10 levels	[34]
Neuroblastoma	➢ CAFs produce pro-inflammatory lipid mediators that induce infiltration of CD163+ macrophages	[35]
Human colorectal cancer	➢ CAFs induce adhesion of monocytes by upregulating VCAM1 levels ➢ CAFs attract monocytes and promote M2 polarization by secretion of IL8 ➢ CAFs and M2 macrophages suppress NK cell function	[37]
Hepatocellular carcinoma	➢ CAFs induce M2 polarization by upregulating mRNA levels of CD206 and CD163 and downregulating mRNA levels of IL-6	[38]

## 5. Macrophages and Virotherapy

Oncolytic virotherapy has become one of the most promising approaches for cancer therapy as demonstrated by the approval of three OVs in clinical setting [4]. Oncolytic virotherapy is based on the use of OVs able to selectively replicate in and lyse tumor cells without harming normal cells. The direct oncolytic activity while promoting tumor burden reduction also induces antigen release, triggering broad innate and acquired immune response conferring to OVs immune modulatory properties. On the other hand, OVs can also activate an anti-viral response by innate immune cells, which limits viral replication and eliminates the OV particles [39], thus preventing virotherapy efficacy. Among these cells, macrophages eliminate viruses in an interferon-dependent manner [40]. However, when viral infection proceeds in macrophages, the effect is an enhanced viral persistence and dissemination. M1 and M2 macrophages show different susceptibilities to virus infection, thus holding capacities to regulate virus loads in the hosts are done through different ways [41]. M2 TAMs act as reservoirs of replication for many viruses: from the human immunodeficiency virus (HIV) to the human cytomegalovirus (HCMV) [42,43]. Recent studies have demonstrated that the vesicular stomatitis virus (VSV), a naturally occurring OV, can infect and replicate only in M2, but not in M1 macrophages [44].

### 5.1. Macrophages in Tumor Response to OVs

In some tumor models, OVs have been shown to induce macrophage infiltration following infection; however, although TAM presence is associated with poor prognosis in cancer patients, in these settings, the recruited macrophages showed increased immuno-stimulatory activity correlating with enhanced and effective anti-tumor response [45]. In a model of glioma, adenoviruses and HSV stimulated the infiltration of macrophages, both in vitro and in vivo [46,47], and these results were corroborated in clinic. The efficacy of virotherapy is closely related to the macrophage phenotype present in the TME. In this context, M1 macrophages reduce tumor growth acting in synergy with OVs; however, they also enhance OV clearance, by the secretion of type 1 IFNs and phagocytosis of OV particles. This double controversial activity is the object of scientific debate since most of the data were obtained in murine models that have limited translational applications, as monocytes in mice differ from those in humans [48]. In contrast, despite their anti-inflammatory effect and the prevention of viral clearance, the pro-tumor effects of M2 macrophages would counteract the anti-tumor effect of OVs. However, several studies demonstrated a suppression of M2 markers upon OV treatment with upregulation of M1 phenotype as described in several cancer models, glioblastoma multiforme murine model, anaplastic thyroid carcinoma xenograft models, breast cancer, and melanoma [48]. Considering the important role played by macrophages in the outcome of virotherapy post-infection, a better understanding of the interactions between OVs and immune cells, particularly M1 and M2 polarized macrophages are needed. A strategy to improve OV efficacy is the recruitment and/or reprogramming of TAMs toward an immune-stimulatory anti-tumor phenotype that could be guided toward tumor cell destruction instead of OV removal [49].

### 5.2. Modulation of Macrophage Phenotype by OVs

Macrophage innate plasticity led to research activities focused on the modulation of TAM phenotype. TAM potential redirection towards an anti-tumor phenotype associated with the M1 profile and phagocytosis would represent a therapeutic advantage. OVs beyond their direct lytic activity on cancer cells, might also redirect TAM phenotype; however, the role of macrophages in OV treatment is dependent on the type of tumor.

In pancreatic cancer models in vivo, the use of an oncolytic vaccinia virus expressing IL-21 (armed oncolytic virus), VVL-21 to infect cancer cells cultured with macrophages, increased the expression of MHCII M1 marker and decreased the expression of CD206 M2 marker. The induction of MHCII marker by VVL21 was also observed in polarized M1 and M2 macrophages. Additionally, VVL-21 infection increased the expression of M1 cytokine gene transcripts (IL-6, IL-12, and COX2) and reduced M2 cytokine gene transcripts (IL-10, TGFβ, or CCL22) in naïve macrophages, in M1 or M2 polarized macrophages [50].

In breast cancer in vitro models, the anti-tumor activity of the oncolytic paramyxoviruses (measles/mumps) was enhanced by human monocyte-derived macrophages, independently of the initial macrophage polarization state and virus replication [51]. In syngeneic, immunocompetent mouse models of primary breast cancer, the OV HSV1716, a modified HSV, reprogrammed TAMs toward a pro-inflammatory and perivascular phenotype, decreasing the prevalence of “tumor promoting” perivascular macrophages. Moreover, HSV1716 significantly increased the number of F4/80 TAM expressing pro-inflammatory, M1-like markers, IL-12 and iNOS, and reduced the expression of the M2-like marker MRC1 [52].

In an anaplastic thyroid carcinoma model, the adenovirus *dl*922-947 induced the switch of tumor macrophages toward a pro-inflammatory M1 phenotype and reduced TAM density in vivo models. TAM depletion in *dl*922-947-treated tumors is associated with increased expression of iNOS and IFN mRNA (encoding the protein IFNγ) [5,53].

In glioblastoma patients treated with the adenovirus Delta24-RGD, tumor macrophages showed a more increased M1 profile than control glioblastoma tissue, indeed, in cerebrospinal fluid samples from treated patients, high pro-inflammatory cytokine levels were detected [54]. In a phase I/II clinical trial, ParvOryx01, a rat H-1 parvovirus (H-1PV) for the treatment of recurrent glioblastoma patients, promoted an immunogenic TME, inducing accumulation of activated TAMs in CD40L-positive glioblastoma regions and up-regulation of cathepsin B and iNOS expression in TAMs [55].

The prominent role of TAMs in cancer progression and modulation of OV therapeutic efficacy has been investigated in several cancer models and strategies aimed at modulating TAM activity are being developed and tested in clinical trials.

## 6. Conclusions

Macrophages are key players in cancer progression and their interaction with cancer/stromal cells contributes to remodel the TME. Additionally, their plasticity makes them able to adopt either pro-inflammatory/anti-tumor or anti-inflammatory/pro-tumor phenotypes in response to TME stimuli. In the context of virotherapy, a growing body of studies describes the ability of OV to modulate macrophage activation status/polarization: the pro-tumor effects of TAMs are reduced as consequence of either enhanced infiltration/depletion or repolarization toward the M1 phenotype. However, an accurate examination of TAM phenotype and their number, pre and post OV infection should be taken into account to better understand the role played by these cells in patients undergoing virotherapy and the mechanisms by which OVs and macrophages interact to yield effective responses.

Among the research areas deserving investigation to provide a deeper understanding of the crosstalk between macrophages and other cells in the TME, TAM origin and differentiation, function, and plasticity as well as mechanisms regulating their activity must be taken into account. Of note, beyond TAM crosstalk with cancer and stromal cells, further exploitation should deal with the interplay with other immune cells, to get a complete analysis of all the factors involved in the TME and how their interplay influence the balance between a hot and a cold tumor. The understanding of all TME components modulating TAM phenotype might address targeted therapies or combinatory approaches aimed at the development of more comprehensive anti-cancer strategies.

## Figures and Tables

**Figure 1 biomedicines-09-01309-f001:**
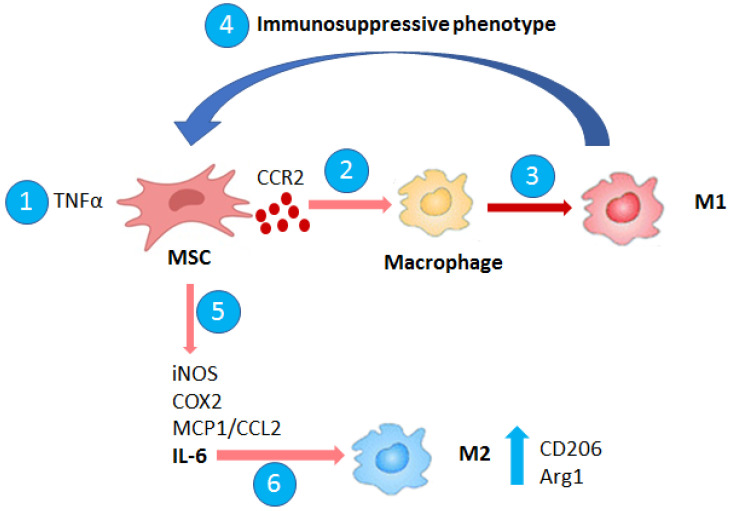
Macrophage–MSC interaction. MSCs activated by TNFα recruit macrophages via CCR2 secretion. M1 macrophages induce MSC immunosuppressive phenotype and enhance iNOS, COX2, MCP1/CCL2, and IL-6 levels. In particular, IL-6 secretion by MSCs induces M2 phenotype with increase of CD206 and ARG1 levels.

## Data Availability

Not applicable.

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
