# Peer review of "Cross Talk of Macrophages with Tumor Microenvironment Cells and Modulation of Macrophages in Cancer by Virotherapy"

_biomedicines, 2021, doi:10.3390/biomedicines9101309_

Round 1
Reviewer 1 Report
In the submitted review “Cross talk of macrophages with tumor microenvironment cells and modulation of macrophages in cancer by virotherapy,” the authors summarize the bidirectional regulation of macrophages by other tumor microenvironment cells like – cancer cells, mesenchymal stem cells, and cancer-associated fibroblasts and the effects of oncolytic virotherapy on macrophage functions. While the authors’ efforts in the summarization are laudable, the review needs further attention in detail at some points. It would be a more comprehensive understanding of the subject if the authors highlighted the specific contribution of each of the molecular parameters described in the polarization of macrophages and thereby regulating the immune response. Some of these points have been noted below.
For the sake of readers’ clarity, it would be beneficial to point out how M1 versus M2 macrophage polarity acts in cancer cell killing or acts in a pro-tumorigenic role, specifically how M2 (molecular cue)-mediated tissue repair functions help in cancer progression.
Line 63-64 – For clarity, it would be appreciable to specify the particular STAT phosphorylation responsible for the M1 polarization within TAM.
Line 67-68 – M2 macrophages are noted to produce and be stimulated by the same set of factors (TGF-b, PGE-2, IL-4/10). Is this information correct?
Line 82-83 – “cancer cells promote macrophage recruitment through the release of chemotactic signals such as VEGFA, CCL2, CXCL12, semaphorin 3A (SEMA3A)’ but it is not clear whether the all macrophages respond to such signals or certain macrophage population specifically responds to such signals.
Line 113 – It would clarify if the authors could expand on how Glycolysis or VACM-1 expression helps in tumor progression.
Line 114 – Lactate dehydrogenase (PDH)?
In section 3, the focus of the paragraph is the cross-talk between MSCs and macrophages. Arguably, the discussion incorporates the presence or the involvement of the cancer cells. Still, it would be easier to follow if the discussion is focused on the effect of macrophage function and how that promotes or antagonizes cancer progression. For example, in line 130, the authors noted that breast cancer cells increase RANTES and CCL5 expression in MSCs, but its direct effects on macrophages and their function were not highlighted further. Similarly, the factors contributing to cancer cell survival signals like IL-6/IL-8/VCAM-1/CCL2 were noted, but their impact on macrophages was not further discussed. Streamlining the review article to keep it focused on the titular topic would enhance the readability of the review article.
Line 183 – Ratio of CAF to TAMs in the tumor microenvironment – Does the stage of cancer progression regulate their relative abundance?
Line 237-239 – How do M1 TAMs help in the clearance of OVs? The adjacent lines are opposing each other in terms of inducing the functional behaviors of OVs. How is the regulation is maintained? Similarly, it would strengthen the article if the effects of M2 TAMs on OV function and clearance could be elaborated with specific examples.
In section 5, ‘macrophages and virotherapy,’ further exploration of all the OVs mentioned in the introduction regarding their association and regulation of macrophages would make the review a more comprehensive read.
Author Response
In the submitted review “Cross talk of macrophages with tumor microenvironment cells and modulation of macrophages in cancer by virotherapy,” the authors summarize the bidirectional regulation of macrophages by other tumor microenvironment cells like – cancer cells, mesenchymal stem cells, and cancer-associated fibroblasts and the effects of oncolytic virotherapy on macrophage functions. While the authors’ efforts in the summarization are laudable, the review needs further attention in detail at some points. It would be a more comprehensive understanding of the subject if the authors highlighted the specific contribution of each of the molecular parameters described in the polarization of macrophages and thereby regulating the immune response. Some of these points have been noted below.
For the sake of readers’ clarity, it would be beneficial to point out how M1 versus M2 macrophage polarity acts in cancer cell killing or acts in a pro-tumorigenic role, specifically how M2 (molecular cue)-mediated tissue repair functions help in cancer progression.
Line 63-64 – For clarity, it would be appreciable to specify the particular STAT phosphorylation responsible for the M1 polarization within TAM.
Authors’ reply
According to the reviewer comment we have provided a brief description of the role of STAT phosphorylation in M1 polarization.
Line 67-68 – M2 macrophages are noted to produce and be stimulated by the same set of factors (TGF-b, PGE-2, IL-4/10). Is this information correct?
Authors’ reply
M2 macrophages as other cells can produce soluble factors that acts in paracrine and autocrine manner as we have described in the text.
Line 82-83 – “cancer cells promote macrophage recruitment through the release of chemotactic signals such as VEGFA, CCL2, CXCL12, semaphorin 3A (SEMA3A)’ but it is not clear whether the all macrophages respond to such signals or certain macrophage population specifically responds to such signals.
Authors’ reply
We have clarified that cancer cells promote monocyte recruitment trough the release of indicated factors contributing to monocyte differentiation into macrophages and polarization of these cells in M1 or M2 TAM depending on TME stimuli.
Line 113 – It would clarify if the authors could expand on how Glycolysis or VACM-1 expression helps in tumor progression.
Authors’ reply
We have better described glycolysis and VCAM1 expression in tumor progression.
Line 114 – Lactate dehydrogenase (PDH)?
Authors’ reply
We have substituted PDH with LDH.
In section 3, the focus of the paragraph is the cross-talk between MSCs and macrophages. Arguably, the discussion incorporates the presence or the involvement of the cancer cells. Still, it would be easier to follow if the discussion is focused on the effect of macrophage function and how that promotes or antagonizes cancer progression. For example, in line 130, the authors noted that breast cancer cells increase RANTES and CCL5 expression in MSCs, but its direct effects on macrophages and their function were not highlighted further. Similarly, the factors contributing to cancer cell survival signals like IL-6/IL-8/VCAM-1/CCL2 were noted, but their impact on macrophages was not further discussed. Streamlining the review article to keep it focused on the titular topic would enhance the readability of the review article.
Authors’ reply
We thank the reviewer for this comment, we had to involve cancer cells at the beginning of the paragraph because they influence MSCs, however as suggested by the reviewer to keep the focus we highlighted in the text the effects and impact on macrophages.
Line 183 – Ratio of CAF to TAMs in the tumor microenvironment – Does the stage of cancer progression regulate their relative abundance?
Authors’ reply
Certainly, the stage of cancer progression is related to CAF and TAM presence in the TME, we have better explained this point in paragraph 4.1.
Line 237-239 – How do M1 TAMs help in the clearance of OVs? The adjacent lines are opposing each other in terms of inducing the functional behaviors of OVs. How is the regulation is maintained? Similarly, it would strengthen the article if the effects of M2 TAMs on OV function and clearance could be elaborated with specific examples.
Authors’ reply
We have better explained that OV treatment promotes a shift from M2 to M1 phenotype, and we have clarified the questions raised by the reviewer in the text.
In section 5, ‘macrophages and virotherapy,’ further exploration of all the OVs mentioned in the introduction regarding their association and regulation of macrophages would make the review a more comprehensive read.
Authors’ reply
Studies reporting regulation of macrophages by the approved OVs are lacking. The only article reporting an enhanced recruitment of immune cells particularly T-cells, NK cells, monocytes and dendritic cells in T-VEC treatment does not deeply investigate monocyte phenotype. (Ramelyte et al, Cancer Cell 39, 394-406 2021)
Reviewer 2 Report
Reviewer’s Comments:
The manuscript "Cross talk of macrophages with tumor microenvironment cells and modulation of macrophages in cancer by virotherapy" by Somma et al discusses the interplay between macrophages and mesenchymal stem cells and macrophages and fibroblasts in tumor microenvironment.
- Please explain the terms “oncolytic virus” and “oncolytic virotherapy” and their use in detail in the introduction section.
- Please rewrite lines 85-91 i.e., “Cancer metabolism ………(HMGB1)”, in section 2.1, so that it is easier to follow.
- Vascular endothelial growth factor is abbreviated as “VEGF”, not as “Vegf”, and Arginase1 is abbreviated as “ARG1”, not as “Arg1”. Please correct these throughout the paper.
- Please specify if the studies being discussed were conducted in vitro, in vivo, or ex vivo.
- In section 5.2, please rephrase the term “re-education”. Please explain what exactly does it mean in this context – it is to differentiate or to redirect? Rewrite section 5.2 so ensure better paragraph flow and clear sentences.
- Please discuss the untapped area of research that needs to be explored to gain a deeper understanding of the interplay between macrophages and other cell types in the tumor microenvironment. This should be added after the conclusions section.
- Please proofread for grammatical and syntax errors.
- Please be consistent with the style of references.
Author Response
The manuscript "Cross talk of macrophages with tumor microenvironment cells and modulation of macrophages in cancer by virotherapy" by Somma et al discusses the interplay between macrophages and mesenchymal stem cells and macrophages and fibroblasts in tumor microenvironment.
- Please explain the terms “oncolytic virus” and “oncolytic virotherapy” and their use in detail in the introduction section.
Authors’ reply
According to the reviewer request, we have explained the terms oncolytic virus and oncolytic virotherapy.
- Please rewrite lines 85-91 i.e., “Cancer metabolism ………(HMGB1)”, in section 2.1, so that it is easier to follow.
Authors’ reply
We have rewritten the sentences to make easier to follow the section 2.1
- Vascular endothelial growth factor is abbreviated as “VEGF”, not as “Vegf”, and Arginase1 is abbreviated as “ARG1”, not as “Arg1”. Please correct these throughout the paper.
Authors’ reply
We have corrected throughout the paper VEGF and ARG1.
- Please specify if the studies being discussed were conducted in vitro, in vivo, or ex vivo.
Authors’ reply
- We have indicated if the experiments were conducted in vitro, in vivo, or ex vivo where not specified.
- In section 5.2, please rephrase the term “re-education”. Please explain what exactly does it mean in this context – it is to differentiate or to redirect? Rewrite section 5.2 so ensure better paragraph flow and clear sentences.
Authors’ reply
The term “re-education” is usually adopted in literature; however, we have substituted it with re-modulation. We have kept the term re-direction since TAM being already differentiated cells can modify their activity based on stimuli received. We have rewritten 5.2 section to make it more clear.
- Please discuss the untapped area of research that needs to be explored to gain a deeper understanding of the interplay between macrophages and other cell types in the tumor microenvironment. This should be added after the conclusions section.
Authors’ reply
As suggested by the reviewer we have further discussed research area dealing with macrophages and other cell types in the TME. This paragraph has been added in the conclusion since no other sections are required after the conclusions by the journal guidelines.
- Please proofread for grammatical and syntax errors.
Authors’ reply
We have proofread grammatical and syntax errors.
- Please be consistent with the style of references.
Authors’ reply
We have used appropriate style of references.
Round 2
Reviewer 1 Report
The authors have addressed the suggestions satisfactorily and the revised manuscript is more focused on the titular subject.